# A Computational Approach to a Model for HIV and the Immune System Interaction

Attaullah [1], Zeeshan [1], Muhammad Tufail Khan [1], Sultan Alyobi [2], Mansour F. Yassen [3,4] and Din Prathumwan [5,*]

[1] Department of Mathematics & Statistics, Bacha Khan University, Charsadda 24461, Pakistan
[2] King Abdulaziz University, College of Science & Arts, Department of Mathematics, Rabigh, Saudi Arabia
[3] Department of Mathematics, College of Science and Humanities in Al-Aflaj, Prince Sattam Bin Abdulaziz University, Al-Aflaj 11912, Saudi Arabia
[4] Department of Mathematics, Faculty of Science, Damietta University, New Damietta 34517, Egypt
[5] Department of Mathematics, Faculty of Science, Khon Kaen University, Khon Kaen 40002, Thailand
* Correspondence: dinpr@kku.ac.th

**Abstract:** This study deals with the numerical solution of the human immunodeficiency virus (HIV) infection model, which is a significant problem for global public health. Acquired immunodeficiency syndrome (AIDS) is a communicable disease, and HIV is the causative agent for AIDS, which damages the ability of the body to fight against disease and easily usual innocuous infections attack the body. On entering the body, HIV infects a large amount of CD4+ T-cells and disturbs the supply rate of these cells from the thymus. Herein, we consider the model with variable source terms in which the production of these cells is a monotonically decreasing function of viral load. Based on the reproduction number, we describe the stability of free equilibrium. The continuous Galerkin–Petrov method, in particular the cGP(2)-method, is implemented to determine the numerical solutions of the model. The influence of different parameters on the population dynamics of healthy/infected CD4+ T-cells and free HIV particles are examined, and the results are presented graphically. On the other hand, the model is solved using the fourth-order Runge–Kutta method, and briefly, the RK4-method, and the results of the proposed schemes are compared with those obtained from other classical schemes such as the Bessel collocation method (BCM), Laplace Adomian decomposition method (LADM), perturbation iteration algorithm (PIA), modified variational iteration method (MVIM), differential transform method (DTM), and exponential Galerkin method (EGM), numerically. Furthermore, absolute errors relative to the RK4 method are computed to describe the accuracy of the proposed scheme. It is presented that the cGP(2)-method gains accurate results at larger time step sizes in comparison with the results of the aforementioned methods. The numerical and graphical comparison reveals that the proposed scheme yields more accurate results relative to other traditional schemes from the literature.

**Keywords:** CD4+ T-cells; immune system; computational analysis

## 1. Introduction

Dynamical systems are well known for their various applications, such as in a population growth model, biomedical, and engineering. For example, in the field of population dynamics, one can study the changes in population with respect to time and make a long-term prediction about population growth for the future. Ding and Ye [1] used a fractional order model of HIV infection. They showed a non-negative solution of that model along with stability analysis. They also expressed numerical simulation and displayed the result of the HIV mathematical model. Burton and Mascola [2] analyzed that AIDS is a result of HIV that minimizes the capability of body scraps beside the disorder and makes it more suitable for natural inoffensive infection. HIV infects a huge quantity of T-cells and divides rapidly while inflowing into the host. Throughout the earliest phases of the disease, blood

consists of the soaring power of HIV harmful particles, which extend throughout the body as the blood circulates. These microorganisms swell into liquid substances of the body, e.g., blood, tears, urine, etc. Most HIV-infected individuals will develop AIDS 10 to 15 years after infection, while several people stay fit for more than this period without ailment. Samanta [3] identified that HIV has a stretched evolution and transmissible phase. In the nonexistence of antiretroviral treatment, the usual period of evolution from HIV to AIDS is 9 to 10 years. The average endurance duration following AIDS is 9 months and 20 days.

Furthermore, the ratio of medical infection evolution depends broadly on personalities, from 14 days until 20 years. Several issues upset the ratio of development. These causes embrace the host's capacity to protect in opposition to HIV. Humble entrance to physical care and the presence of associated infections may also inspire quicker illness development. Krischner et al. [4] discussed treating HIV infection and certain chemotherapies have been tested. The most commonly used drugs are AZT, DDI, DDC, and D4T. All these drugs are inhibiters, which work like reverse transcriptase inhibitors. The reverse transcriptase inhibitors reduce the growth of the virus and finally block the infection.

Many researchers have worked in this field to conduct studies and analyze research to control this disease. Chum et al. [5] addressed the rule of sexually transferred infection in HIV-1. Sun and Min [6] studied a modified HIV infection model simulation. Khalil et al. [7] analyzed HIV infection with the drug therapy effect. They also discussed the generalized Euler method (GEM) and the numerical solution of the HIV fractional-order model. Wang et al. [8] examined the HIV pathogenesis model along with its cure rate and logistic growth of healthy and infected CD4+ T-cells. Osman and Abdurrahman [9] studied the stability of the delayed HIV model and investigated its treatment and transmission. They also computed the basic reproduction number $R_0$ and discussed its stability based on the reproduction number. Wan et al. [10] discussed the behavior of the viral dynamical model and its solution. Wang et al. [11] addressed the global dynamics of HIV infection of CD4+ T-cells and analyzed the mathematical model. Srivastava et al. [12] presented drug therapy for HIV infection. Culshaw et al. [13] demonstrated the differential equation of HIV infection and discussed the modeling and dynamics of CD4+ T-cells.

HIV/AIDS is a universal health challenge; over 70 million people worldwide have been infected with this disease, from which 35 million people have passed away, and 36.7 million patients are still living with this infection [14]. Nelson and Perelson [15] worked on the model of HIV infection and its mathematical analysis based on kinetic parameters. They analyzed intracellular delays, antiretroviral therapy, and dynamics of infected and uninfected CD4+ T-cells. They also illustrated that when the drug effect is less than the loss rate of productively infected cells, the approximated value is ideal and depicts the results of the model. Perelson et al. [16] examined the interaction of HIV and CD4+ T-cells. They discussed the infected, active infected, uninfected, and free virus T-cells. They explained the characteristics of HIV infection in a straightforward manner. They considered two types of HIV models. The first is with a constant source for the growth of CD4+ T-cells, and the second is a decrease in the relative viral load of CD4+ T-cells. Ronga et al. [17] discussed a mathematical model to examine the method of drug resistance during therapy and its appearance. They indicated that drug-resistant viruses contributed significantly to their stability. When antiretroviral treatments are available, both strain and wild-type reproduction are close to each other. Duffin and Tullis [18] worked on a mathematical model for HIV infection and AIDS to understand these diseases. They suggested a mathematical model of targeted cells; for this purpose, they evaluated two models and compared the results with actual observations. Song and Chang [19] described the time between the infection of T-cells and the secretion of viral elements. In addition, they demonstrated the condition under which the model is stable and the model's existence in that solution. Mechee and Haitha [20] investigated the application of Lie symmetry for an HIV-infected model that deals with the initial value of a nonlinear differential equation. Based on Lie symmetry approach, Mechee and Haitha [20] attempted to identify uninfected T-cells in the host body. Zhou et al. [21] and Leenheer et al. [22] studied a model for HIV

infection with its cure rate. They proved that when the basic reproduction number, $R_0 < 1$, there are no infections in the host cells, were as when $R_0 > 1$, the host cell is infected by HIV infection. They also found conditions for which the system is stable or unstable. Sarivastav and Chandra [23] proposed the dynamics of HIV and CD4+ T-cells in primary infection, also modeling for HIV infection. They analyzed stability during the infection state of the host cells of the body. They further discussed the local stability of the model and showed the results numerically. They determined the basic reproduction number R0 and discussed that when $R_0$, the host cells are free of disease; when $R_0 > 1$, the host cells are infected with the disease. Liu and Li [24] formulated an HIV infection model that depends on the age of infection, elapsed age of disease, and antiretroviral treatment. The model is concerned with two classes, infective (concentration of infection at birth) and AIDS (concentration of disease without birth), whose results give an approximate interval of optimal pulse and proportion of impulsivity. Ho et al. [25] discussed the rapid turnover of plasma virion and HIV-1 infection. They also demonstrated the treatment of individuals infected with HIV-1 infection, which causes AIDS. Perelson et al. [26] discussed the decay characteristics of HIV-1-infected compartments during combination therapy. Kuang [27,28] proposed the delay differential equation with applications in population dynamics. Ongun [29] implemented the Laplace Adomian decomposition method for solving a model for HIV infection of CD4+ T-cells. Yuzbasi [30] employed the numerical approach to solve the model for HIV infection of CD4+ T-cells. Khalid et al. [31] discussed the numerical solution of a model for HIV infection of CD4+ T-Cells. Merdan et al. [32] described the numerical solution of the model for HIV infection of CD4+ T-Cells. Attaullah et al. [33] studied the transmission and dynamical behavior of an HIV/AIDS epidemic model with a cure rate based on mathematical model. Ogunlaran and Noutchie [34] considered the HIV-infected model with two variables, uninfected CD4+ T-cells and incident term of the free virion. Ogunlaran and Noutchie [34] aimed to increase the concentration of uninfected CD4+ T-cells using minimal drug therapies and to stop the reproduction of infected cells. Boukari et al. [35] analyzed a discrete HIV infection model with a time delay and expressed the global stability of the model numerically. They used the backward Euler method and proved that $E_0$ is globally asymptotically stable when $R_0 < 1$. Li and Xiao [36] investigated the global dynamics of a virus' immune system in order to establish the HIV load and structured treatment outages. They also addressed the global dynamics of the HIV elimination and infection cell growth model. Espindola et al. [37] investigated macrophages and their role in HIV infection. They also discussed how highly active antiretroviral therapy (HAART) affected HIV infection. Kinner et al. [38] examined the incidence of HIV, hepatitis B, and hepatitis C in older and younger adults and deduced that the incidence is lower in younger adults than in older adults. Angulo et al. [39] demonstrated that the main path of HIV-1 infection is transmitted from a mother to her child. They also found polymorphisms in human leukocyte antigen class-B (HLA-B) concerned with HIV-1 infection. Theys et al. [40] studied HIV-1 impact on host cells and their transmission. They also found a link between the evolution of the host cell and the fitness of the host cell. Hallberge et al. [41] established a developed stage of knowledge on the significance of HIV revelation between partners. They discussed that most HIV infection is transferred from one infected person (female) to another person (male) and the status of both are the main factor of this disease. Ransome et al. [42] analyzed the spread, cure, and prevention of HIV infection in social relationships. They realized that social capital is an important factor in HIV transmission from one person to another. Naidoo et al. [43] studied the care of tuberculosis (TB) and their class in those people that are already infected with HIV. Omondi et al. [44] considered a mathematical model of HIV infection and investigated the transmission between two kinds of different ages. They also showed that males are less infected than their female partners by this infection. Duro et al. [45] illustrated the CD4+ T-cell monitoring in HIV-infected people with the help of CD4+ T-cell counts. They also found the possibility of CD4+ T-cells being maintained during viral suppression by using the Kaplan–Meier technique. Mbogo et al. [46] considered the model of HIV infection with the help of the Stochastic approach and the probability

of HIV when clear, which depends on drug cure rate and intracellular delay length, both of which play a vital role in HIV progression. Ghoreishi et al. [47] utilized the homotopy analysis method (HAM) to determine the solution of the HIV-infected model in the form of an infinite series. They used auxiliary parameters to adjust and control the convergence portion of the infinite series. Elaiw [48] presented the HIV model along with two categories of targeted cells, T-cells and macrophages, with an infection rate. They used Lyapunov and Lasalle principles to recognize the global stability of the infected and uninfected state. They computed the basic reproduction number "$R_0$" and analyzed that when "$R_0 < 1$" the uninfected state is globally asymptotically stable, and when "$R_0 > 1$" the infected state is globally asymptotically stable. Ali et al. [49] computed the solution of the HIV-infected model by using the Adomian decomposition method (ADM) that illustrates the solution of ODE's in terms of infinite series components. Yuzbasi and Karacayir [50] considered a model of HIV infection and determined the solution of the model by using the exponential Galerkin method (EGM). They used a technique of residual correction. The purpose of this technique is to reduce the error of the solution. They also showed his result numerically and compared them with numerous existing methods. Kirschner [51] studied HIV immunological dynamics employing mathematics. In the Chemotherapy of AIDS, Webb et al. [52] described the mathematical model for HIV treatment approach. Attaullah and Sohaib [53] implemented two numerical schemes, namely, continuous Galerkin–Petrov (cGP(2)) and Legendre wavelet collocation method (LWCM), for the approximate solution of the mathematical model which describes the behavior of CD4+ T-cells, infected CD4+ T-cells, and free HIV virus particles after HIV infection. They presented and analyzed the effect of constant and different variable source terms (depending on the viral load) used for the supply of new CD4+ T-cells from the thymus on the dynamics of CD4+ T-cells, infected CD4+ T-cells, and free HIV virus. Furthermore, they also solved the model using the fourth-order Runge–Kutta (RK4) method. They highlight the accuracy and efficiency of the proposed schemes with the other traditional schemes.

## 2. Main Objectives

The main contribution of this paper is to present the implementation and analysis of the cGP(2)-method [53] on the HIV-infection model proposed by Yuzbasi and Karacayir [50] and compare the results with those obtained from other conventional methods existing in the literature. Comparisons verify that the new findings align well with the existing solutions. Since the method mentioned earlier is based on the finite element method, the present scheme can be utilized as an adaptive scheme. Moreover, this variational type of discretization has several advantages over the standard schemes. In the existing literature, the majority of the researchers presumed that the supply rate of new CD4+ T cells from the thymus is constant when assessing the dynamics of HIV. However, HIV is capable of infecting these cells in the thymus, and variable phenomena have been observed as contrasted to constant versions. Therefore, we extended the model considered in [50] by introducing a variable source term, a monotonically decreasing function depending on the concentration of viral load. To understand the stability, we determined the basic reproduction number of the model. The model is essential for mathematically simulating HIV infection of CD4+ T cells. This will be employed to study CD4+ T-cell population dynamics in the involvement and complete lack of HIV, which will assist in the perception of diagnostic and therapeutic AIDS symptoms as well as diminishing the disease. This will be a helpful addition to the current literature on biomathematics. In the remainder of the present study, healthy cells, infected cells, and the virus will be used to mean healthy, infected CD4+ T-cells, and free HIV particles, respectively.

## 3. Mathematical Formulation of HIV Model

This section is concerned with the HIV infection model described by Yuzbasi and Karacayir [50], in which the total population is divided into three compartments, i.e., con-

centration of healthy T-cells $T(t)$, concentration of infected T-cells $I(t)$, and concentration of free HIV $V(t)$ at time $t$. The description of the model is as follows:

$$
\begin{aligned}
\frac{dT(t)}{dt} &= \eta - \alpha_T T(t) + \beta_T T(t)\left(1 - \frac{T(t)+I(t)}{T_{\max}}\right) - \psi V(t)T(t), \\
\frac{dI(t)}{dt} &= \psi V(t)T(t) - \phi I(t), \\
\frac{dV(t)}{dt} &= r_I \phi I(t) - \alpha_V V(t).
\end{aligned}
\tag{1}
$$

Initial conditions for this system are $T(0) = T_0$, $I(0) = I_0$, and $V(0) = V_0$, such that $T(t)$ indicates the concentration of healthy T-cells, $I(t)$ indicates the population of infected T-cells, and $V(t)$ indicates the dynamics of free HIV at time $t$.

## 4. Modified Formulation of HIV Model

This section is concerned with the extension of the model (Yuzbasi and Karacayir [50]) by introducing a variable source term depending on the viral load instead of using a constant source for the supply of new cells from the thymus, i.e., "$0.5\eta + \frac{5\eta}{1+V(t)}$" [51,52] where "$\eta$" is the supply rate of healthy cells. The modified model is as follows:

$$
\frac{dT(t)}{dt} = 0.5\eta + \frac{5\eta}{1+V(t)} - \alpha_T T(t) + \beta_T T(t)\left(1 - \frac{T(t)+I(t)}{T_{max}}\right) - \psi V(t)T(t), \tag{2}
$$

$$
\frac{dI(t)}{dt} = \psi V(t)T(t) - \phi I(t), \tag{3}
$$

$$
\frac{dV(t)}{dt} = r_I \phi I(t) - \alpha_V V(t). \tag{4}
$$

The initial conditions and parameters involved in the model are explained in Table 1.

**Table 1.** The explanation of parameters with their values (Yuzbasi and Karacayir [50]).

| Variables | Description | Values |
|:---:|:---:|:---:|
| $T_0$ | Concentration of healthy cells | 0.1 mm$^{-3}$ |
| $I_0$ | Population of infected cells | 0.0 mm$^{-3}$ |
| $V_0$ | Dynamics of free viruses | 0.1 day$^{-1}$mm$^{-3}$ |
| $\eta$ | Supply rate of healthy cells | 0.1 day$^{-1}$mm$^{-3}$ |
| $\alpha_T$ | Natural death rate for healthy cells | 0.02 day$^{-1}$ |
| $T_{max}$ | Maximum density of healthy cells population | 1500 day$^{-1}$mm$^{-3}$ |
| $\psi$ | Infection rate of healthy cells | 0.0027 day$^{-1}$ |
| $\phi$ | Virus particles released by infected cells | 0.3 day$^{-1}$ |
| $\alpha_V$ | Virus death rate | 2.4 day$^{-1}$ |
| $r_I$ | Death rate of infected cells | 10 mm$^{-3}$ |
| $\beta_T$ | Growth rate of healthy cells | 3 day$^{-1}$ |

### 4.1. Uninfected Steady State

In this condition, viruses are not present in the human cells and in other living things. In this situation, "$E_0$" is stable and "$\bar{E}$" is unstable in the uninfected steady state. In the uninfected steady condition, we consider that $T = T_0$, $I = 0$, and $V = 0$. Then, we discover "$T_0$" from the uninfected condition from System (1), which after some simplification, is as follows:

$$
\left(\frac{\beta_T}{T_{max}}\right) T_0(t)^2 + (\beta_T - \alpha_T) T_0(t) - 5.5\eta = 0,
$$

then, using the quadratic formula to find the value of "$T_0$" we have:

$$
T_0 = \frac{-B \pm \sqrt{B^2 - 4AC}}{2A},
$$

where $A = \frac{\beta_T}{T_{max}}$, $B = (\beta_T - \alpha_T)$, and $C = -5.5\eta$.

Using the above we get:

$$T_0 = \frac{T_{max}}{2\beta_T}\left[(\alpha_T - \beta_T) \pm \sqrt{(\beta_T - \alpha_T)^2 - \frac{22\beta_T\eta}{T_{max}}}\,\right].$$

### 4.2. Infected Steady State

In an infected state, viruses are in human cells and in other living things. In an infected steady state, we consider that $T = \overline{T}$, $I = \overline{I}$, and $V = \overline{V}$ in System (1). To obtain the value of $\overline{T}$, $\overline{I}$, and $\overline{V}$, we get the following after simplification:

$$\overline{T}(t) = \frac{\alpha_V}{\psi r_I}$$

$$\overline{I}(t) = \frac{T_{max}\psi r_I\left(\eta - \alpha_T\overline{T}(t) - \alpha_T\overline{T}(t)V(t)\right)}{(\beta_T + V(t))\left(\alpha_V - \overline{T}(t)\alpha_V - \psi r_I\phi T_{max}\right)}$$

$$\overline{V}(t) = \frac{\phi r_I}{\alpha_V}\,\overline{I}(t)$$

### 4.3. Reproduction Number

In the cure rate of infection, the virus can easily be controlled when the value of the reproduction number is smaller because the viral infection speed is less. However, the disease cannot be controlled easily without a cure rate because the viral infection speed is larger, which shows a faster speed of the virus. We determined that the reproduction number $R_0$, and observe that the local and global dynamics are absolutely resolute by the significance of $R_0$. When $R_0 \leq 1$, then the disease-free equilibrium "$E_0$" is locally and globally asymptotically stable, while $R_0 > 1$, then the disease-free equilibrium "$E_0$" becomes locally and globally asymptotically unstable. The reproduction number is $R_0 = \frac{T_0}{\overline{T}}$, where $\overline{T}(t) = \frac{\alpha_V}{\psi r_I}$.

### 4.4. Jacobian Matrix

The Jacobian matrix corresponding to System (7) about $E(T, I, V)$ is provided by:

$$J(E) = \begin{pmatrix} -\alpha_T + \beta_T - \psi V - \frac{2\beta_T T + \beta_T I}{T_{max}} & -\frac{\beta_T T}{T_{max}} & -\frac{5\eta}{(1+V(t))^2} - \psi T \\ \psi V & -\phi & \psi T \\ 0 & r_I\phi & -\alpha_V \end{pmatrix}$$

### 4.5. Stability Analysis

The variational matrix of System (1) about $E_0(M, 0, 0)$ is provided as follows:

$$J(E_0) = \begin{pmatrix} -\alpha_T + \beta_T - \frac{2\beta_T M}{T_{max}} & -\frac{\beta_T M}{T_{max}} & -5\eta - \psi M \\ 0 & -\phi & \psi M \\ 0 & r_I\phi & -\alpha_V \end{pmatrix}.$$

**Theorem 1.** *If $R_0 \leq 1$, the disease-free equilibrium $E_0(M, 0, 0)$ is locally and globally asymptotically stable.*

**Proof.** The Jacobian matrix for the given System (1) about $E_0(M, 0, 0)$ is as follows:

$$J(E_0) = \begin{pmatrix} -\alpha_T + \beta_T - \frac{2\beta_T M}{T_{max}} & -\frac{\beta_T M}{T_{max}} & -5\eta - \psi M \\ 0 & -\phi & \psi M \\ 0 & r_I\phi & -\alpha_V \end{pmatrix},$$

the characteristic equation of the above matrix is $\det(\lambda I - J(E_0)) = 0$, where "$I$" is the unit matrix. Then, to expand this matrix with respect to the 1st column we obtained

$$\left(\lambda + \alpha_T - \beta_T + \frac{2\beta_T M}{T_{max}}\right)\left(\lambda^2 + \alpha_V \lambda + \phi\lambda + \phi\alpha_V - \phi\psi r_I M\right) = 0,$$

$$\left(\lambda + \alpha_T - \beta_T + \frac{2\beta_T M}{T_{max}}\right)\left(\lambda^2 + (\alpha_V + \phi)\lambda + \phi\alpha_V - \phi\psi r_I M\right) = 0,$$

$$(\lambda + b)\left(\lambda^2 + a_1\lambda + a_0\right) = 0,$$

where $a_1 = \alpha_V + \phi$,

$$a_0 = \phi\alpha_V - \phi\psi r_I M,$$

and

$$b = \alpha_T - \beta_T + \frac{2\beta_T M}{T_{max}}.$$

It is clear from the above equation that one root of this equation is $\lambda = -b$. Then, to determine the other roots, we consider the equation as:

$$\lambda^2 + a_1\lambda + a_0 = 0,$$

when $R_0 < 1$, $a_1 > 0$, $a_0 > 0$, and $b > 0$.

Hence, by Hurwitz Criterion [28] all roots of the given equation have a negative real part. □

**Theorem 2.** *The disease-free equilibrium $E_0(M, 0, 0)$ is locally and globally asymptotically unstable when $R_0 > 1$.*

**Proof.** $E_0(M, 0, 0)$, given from the Jacobian matrix of System (7), is presented below:

$$J(E_0) = \begin{pmatrix} -\alpha_T + \beta_T - \frac{2\beta_T M}{T_{max}} & -\frac{\beta_T M}{T_{max}} & -5\eta - \psi M \\ 0 & -\phi & \psi M \\ 0 & r_I\phi & -\alpha_V \end{pmatrix},$$

the characteristic equation of the above matrix as $\det(\lambda I - J(E_0)) = 0$, and "$I$" is the unit matrix. Then, we expand this matrix with respect to the 1st column and obtained:

$$\left(\lambda + \alpha_T - \beta_T + \frac{2\beta_T M}{T_{max}}\right)\left(\lambda^2 + \alpha_V \lambda + \phi\lambda + \phi\alpha_V - \phi\psi r_I M\right) = 0,$$

$$\left(\lambda + \alpha_T - \beta_T + \frac{2\beta_T M}{T_{max}}\right)\left(\lambda^2 + (\alpha_V + \phi)\lambda + \phi\alpha_V - \phi\psi r_I M\right) = 0,$$

$$(\lambda + b)\left(\lambda^2 + a_1\lambda + a_0\right) = 0,$$

where $a_1 = \alpha_V + \phi$,

$$a_0 = \phi\alpha_V - \phi\psi r_I M,$$

and $b = \alpha_T - \beta_T + \frac{2\beta_T M}{T_{max}}$.

It is clear from the above equation that one root of this equation is $\lambda = -b$. Then, to determine the other roots, we consider the equation:

$$\lambda^2 + a_1\lambda + a_0 = 0,$$

when $R_0 > 1$, $a_1 > 0$, $a_0 > 0$, and $b > 0$.

Hence by Hurwitz-Criterion [28] all roots of this scheme have a positive real part. □

**Theorem 3.** *For any positive solution* $(T(t), I(t), V(t))$ *of System (1), there is* $N > 0$*, such that* $T(t) \leq N$*,* $I(t) \leq N$*, and* $V(t) \leq N$*,* $\forall$ *large t.*

**Proof.** Let

$$L(t) = T(t) + I(t), \tag{5}$$

taking the derivative of Equation (5), with the solution of System (1), to obtain:

$$L'(t) = T'(t) + I'(t),$$

$$L'(t) = 0.5\eta + \frac{5\eta}{1 + V(t)} - \alpha_T T(t) + \beta_T T(t) \left(1 - \frac{T(t) + I(t)}{T_{max}}\right) - \psi V(t)T(t) + \psi V(t)T(t) - \phi I(t),$$

$$L'(t) = 0.5\eta + \frac{5\eta}{1 + V(t)} - \alpha_T T(t) + \beta_T T(t) \left(1 - \frac{T(t) + I(t)}{T_{max}}\right) - \phi I(t),$$

$$L'(t) = 0.5\eta + \frac{5\eta}{1 + V(t)} - \alpha_T T(t) + \beta_T T(t) + \frac{-\beta_T T(t)^2 + \beta_T T(t)I(t)}{T_{max}} - \phi I(t), \leq -hL(t) + N_0,$$

where $N_0 = (T_{max}\beta_T{}^2 + 4\beta_T\eta)/4\beta_T$, $h = \min(\alpha_T, \phi)$.

Then, $N_1 > 0$ *exists*, depending on the parameter of the given system. Therefore, $L(t) < N_1$, for all $t$. Then, $T(t)$ and $I(t)$ are both bounded above. Then, we also have $V(t)$ bounded above from the 3rd equation of System (1), and the maximum of this is $N$. The proof is complete. $\square$

**Theorem 4.** Suppose that

(a) $R_0 > 1$,

(b)
$$\left(\alpha_V + \phi - u + \frac{2\beta_T T(t)}{T_{max}}\right)\left(\phi\alpha_V - \phi r_I\psi\overline{T}(t) - u\alpha_V - u\phi + \frac{2\beta_T T(t)\alpha_V}{T_{max}} + \right.$$
$$\frac{2\beta_T \overline{T}(t)\phi}{T_{max}} + \frac{\psi\overline{V}(t)\beta_T\overline{T}(t)}{T_{max}}\right)\left(-u\phi\alpha_V + ur_I\phi\psi\overline{T}(t) + \frac{2\beta_T\overline{T}(t)\phi\alpha_V}{T_{max}} - \right.$$
$$\left.\frac{2\beta_T\overline{T}(t)^2 r_I\phi\psi}{T_{max}} + \frac{\psi\overline{V}(t)\alpha_V\beta_T\overline{T}(t)}{T_{max}} + \psi^2 r_I\phi\overline{T}(t)\overline{V}(t) + \frac{5\eta\psi r_I\phi\overline{V}(t)}{(1+\overline{V}(t))^2}\right) > 0$$

*Then, the equilibrium* $\overline{E}(\overline{T},\ \overline{I},\ \overline{V})$ *is locally asymptotically stable.*

**Proof.** For equilibrium $\overline{E}(\overline{T},\ \overline{I},\ \overline{V})$, the given system after simplification reduces to

$$\lambda^3 + a_2\lambda^2 + a_1\lambda + a_0 = 0,$$

where,

$$a_2 = \alpha_V + \phi - u + \frac{2\beta_T\overline{T}(t)}{T_{max}} > 0,$$

$$a_1 = \phi\alpha_V - \phi r_I\psi\overline{T}(t) - u\alpha_V - u\phi + \frac{2\beta_T\overline{T}(t)\alpha_V}{T_{max}} + \frac{2\beta_T\overline{T}(t)\phi}{T_{max}} + \frac{\psi\beta_T\overline{V}(t)\overline{T}(t)}{T_{max}} > 0$$

$$a_0 = -u\phi\alpha_V + ur_I\phi\psi\overline{T}(t) + \frac{2\beta_T\overline{T}(t)\phi\alpha_V}{T_{max}} - \frac{2\beta_T\overline{T}(t)^2 r_I\phi\psi}{T_{max}} + \frac{\psi\overline{V}(t)\alpha_V\beta_T\overline{T}(t)}{T_{max}} + \psi^2 r_I\phi\overline{T}(t)\overline{V}(t) + \frac{5\eta\psi r_I\phi\overline{V}(t)}{(1+\overline{V}(t))^2} > 0,$$

and $u = -\alpha_T + \beta_T - \psi\overline{V}(t) - \frac{\beta_T\overline{I}(t)}{T_{max}}$.

To find the second part of this Theorem 4, we have

$$
a_2a_1 - a_0 = \mu\left[\left(\phi\alpha_V - \phi r_I\psi\overline{T}(t) - u\alpha_V - u\phi + \frac{2\beta_T\overline{T}(t)\alpha_V}{T_{\max}} + \frac{2\beta_T\overline{T}(t)\phi}{T_{\max}}\right.\right.
$$
$$
\left.+\frac{\psi\overline{V}(t)\beta_T\overline{T}(t)}{T_{\max}}\right)\left(-u\phi\alpha_V + ur_I\phi\psi\overline{T}(t) + \frac{2\beta_T\overline{T}(t)\phi\alpha_V}{T_{\max}} - \frac{2\beta_T\overline{T}(t)^2 r_I\phi\psi}{T_{\max}}\right.
$$
$$
\left.\left.+\frac{\psi V(t)\alpha_V\beta_T\overline{T}(t)}{T_{\max}} + \psi^2 r_I\phi\overline{T}(t)\overline{V}(t) + \frac{5\eta\psi r_I\phi\overline{V}(t)}{(1+\overline{V}(t))^2}\right)\right] > 0,
$$

where $\mu = \left(\alpha_V + \phi - u + \frac{2\beta_T\overline{T}(t)}{T_{max}}\right)$.

By Routh–Hurwitz criterion [28], i.e.,

$$
a_0 > 0, \ a_1 > 0, \ a_2 > 0,
$$

we also have $a_2a_1 - a_0 > 0$.

Then, $\overline{E}(\overline{T}, \ \overline{I}, \ \overline{V})$ is locally asymptotically stable (LAS) □.

**Theorem 5.** *System (1) is a competitive system.*

**Proof.** Consider a matrix "$R$" and Jacobian matrix of System (1), such that

$$
R = \begin{pmatrix} 1 & 0 & 0 \\ 0 & -1 & 0 \\ 0 & 0 & 1 \end{pmatrix},
$$

System (1) is competitive in "$D$", where $D = (T, I, V) \in R^3 : 0 < T \leq M$, $0 < I \leq M$, $0 < V \leq M$. For some partial order defined as $K = (T, I, V) \in R^3 : T \leq 0$, $I \geq 0$, $V \geq 0$. After simplification, we obtained:

$$
R\frac{\partial f}{\partial x}R = \begin{pmatrix} 1 & 0 & 0 \\ 0 & -1 & 0 \\ 0 & 0 & 1 \end{pmatrix}\begin{pmatrix} -\alpha_T + \beta_T - \psi V - \frac{2\beta_T T + \beta_T I}{T_{max}} & -\frac{\beta_T T}{T_{max}} & -\frac{5\eta}{1+V(t)} - \psi T \\ \psi V & -\phi & \psi T \\ 0 & r_I\phi & -\alpha_V \end{pmatrix}\begin{pmatrix} 1 & 0 & 0 \\ 0 & -1 & 0 \\ 0 & 0 & 1 \end{pmatrix},
$$
$$
R\frac{\partial f}{\partial x}R = \begin{pmatrix} -\alpha_T + \beta_T - \psi V - \frac{2\beta_T T + \beta_T I}{T_{max}} & \frac{\beta_T T}{T_{max}} & -\frac{5\eta}{1+V(t)} - \psi T \\ -\psi V & -\phi & -\psi T \\ 0 & -r_I\phi & -\alpha_V \end{pmatrix}.
$$

This proof is complete. □

## 5. The Numerical Methods

### 5.1. The Continuous Galerkin–Petrov Method

Numerical methods are widely used to simulate complex real-world problems. Jiwari [54–56] proposed the local radial basis function-finite difference based algorithms for the singularly perturbed Burgers' model. Mittal et al. [57] suggested a cubic B-spline quasi-interpolation algorithm to capture the pattern formation of coupled reaction–diffusion models. Pandit [58] discussed the local radial basis functions and scale-3 Haar wavelet operational matrices based on numerical algorithms for a generalized regularized long wave model. Mittal et al. [59] considered a new scale-3 haar wavelet algorithm for numerical simulation of second-order ordinary differential equations. Nowadays, the cGP-method has been successfully employed to solve many types of non-linear problems in science and engineering, for example [27,33–39]. In this paper, we applied this approach to the HIV infection model. The system of ODEs for HIV Model (1) can be considered as:

Find $u : [0, T] \to V = R^d$ such that

$$u'(t) = F(t, u(t)) \; \forall \, t \in (0, T),$$

$$u(0) = u_o, \tag{6}$$

where $u(t) = [T(t), I(t), V(t)]$ and $F$ is the nonlinear right-hand side vector-valued function. At $t = 0$, $u_1 = T(0)$, $u_2 = I(0)$, $u_3 = V(0)$ where $T(0)$, $I(0)$ and $V(0)$ are the initial conditions given in Table 1.

In order to find the approximate solution of System (1), we partitioned the time interval $I = [0, T]$ into a number of small pieces $I_m = (t_{m-1}, t_m]$, where $m = 1, \dots, N$ and $0 = t_o < t_1 \dots < t_{N-1} < t_N = T$.

The symbol $\tau = t_m - t_{m-1}$ is used to represent the maximum time step size. For the derivation of the cGP-method, the system of equations in System (1) is multiplied with suitable test functions (see [33–35,39] for more details) and integrated over $I_m$. The discrete solution $u_1 / I_m$ can be represented by the polynomial ansatz

$$u_\tau(t) / s_m(t) = \sum_{j=o}^{k} U_m{}^j \varphi_{m, j}(t), \tag{7}$$

where $U_m{}^j$ are the members of the function space $V$ and the basis functions $\varphi_{m, j} \in \mathbb{P}_k(I_m)$ are chosen as Lagrange basis functions w. r. t. the $k + 1$ points $t_{m, j} \in I_m$ with the following assumption

$$\varphi_{m, j}(t_{m, i}) = \delta_{i, j}, \; i, \, j = 0, \dots, k, \tag{8}$$

where $\delta_{i, j}$ the usual Kronecker delta. We choose the points as $t_{m, 0} = t_{m-1}$ and $t_{m, 1} = t_m$ the $(k + 1)$-quadrature points of the Gauß–Lobatto formula on each time interval. In this way, the initial condition can be written as

$$U_m{}^0 = u_\tau / s_{m-1}(t_{m-1}), \text{ if } m \geq 2 \text{ or } U_m{}^0 = u_0, \text{ if } m = 1. \tag{9}$$

The basic functions $\varphi_{m, j} \in \mathbb{P}_k(s_m)$ of (7) are defined using the reference transformations (see [33–35,39] for more details). Similarly, the test basis functions $\hat{\psi}_i \in \mathbb{P}_{k-1}(\hat{s})$ are defined with the appropriate choice in order to compute the coefficients (see [33–35,39] for details). Finally, the cGP($k$)-method reads:

$$\sum_{j=0}^{k} \alpha_{i, j} U_m{}^j = \frac{\tau_m}{2} \left\{ F\left(t_{m, i}, U_m{}^i\right) + \beta_i F\left(t_{m, 0}, U_m{}^0\right) \right\}, \; \forall \, i = 1, \dots, k. \tag{10}$$

where $U_m{}^0 = U_{m-1}{}^k$ for $m > 1$ and $U_1{}^0 = u_0$ for $n = 1$ are the initial values $\alpha_{i, j}$ and $\beta_i$ and are defined as:

$$\alpha_{i, j} = \hat{\varphi}'_j(\hat{t}_i) + \beta_i \hat{\varphi}'_j(\hat{t}_0) \text{ and } \beta_i = \hat{E}_0 \hat{\psi}_i(\hat{t}_0), \tag{11}$$

Once the above system is solved, the initial condition for the next time interval $I^-{}_{m+1}$ is set to $U_{m+1}{}^0 = U_m{}^k$. For $k = 2$, the coefficients $\alpha_{i, j}$ and $\beta_{i, j}$ of the cGP(2)-method are computed as follows:

The cGP(2) Method

We used the three-point Gauß–Lobatto formula (Simpson rule) to define the quadratic basis functions with weights $\hat{E}_0 = \hat{E}_2 = 1/3$, $\hat{E}_1 = 4/3$, and $\hat{t}_0 = -1$, $\hat{t}_2 = 0$, $\hat{t}_0 = 1$. Then, we get $(\alpha_{i, j}) = \begin{pmatrix} -\frac{5}{4} & 1 & \frac{1}{4} \\ 2 & -4 & 2 \end{pmatrix}$, $(\beta_i) = \begin{pmatrix} \frac{1}{2} \\ -1 \end{pmatrix}$, $i = 1, 2$ and $j = 0, 1, 2$.

Thus, the system to be solved for $U_m{}^1$, $U_m{}^2 \in V$ from the known $U_m{}^0 = U_{m-1}{}^2$ becomes:

$$\alpha_{1,1}U_m{}^1 + \alpha_{1,2}U_m{}^2 = -\alpha_{1,0}U_m{}^0 + \frac{\tau_m}{2}\left\{F\left(t_{m,1},\ U_m{}^1\right) + \beta_1 F\left(t_{m,0},\ U_m{}^0\right)\right\}, \quad (12)$$

$$\alpha_{2,1}U_m{}^1 + \alpha_{2,2}U_m{}^2 = -\alpha_{2,0}U_m{}^0 + \frac{\tau_m}{2}\left\{F\left(t_{m,2},\ U_m{}^2\right) + \beta_2 F\left(t_{m,0},\ U_m{}^0\right)\right\}, \quad (13)$$

where $U_m{}^0$ represent the initial condition at the current time interval.

### 5.2. The Classical Explicit Runge–Kutta Method

This the fourth order Runge–Kutta method is very well known and was developed by Kutta [54] (see [55] for more details).

Numerical Results and Discussions

In this section, we investigate the numerical solutions of the extended HIV Model (1) by using the cGP(2) method. Initial values of the variables, different parameters, and their detailed descriptions are provided in Table 1. Various parameters present in the model are investigated with the help of graphs to understand their behavior. Figures 1–3 depict the healthy cells, infected cells, and virus particles for different values of the growth rate of healthy cells "$\beta_T$", respectively. It is concluded that the concentration of healthy cells decreases, whereas the distribution of infected cells and the virus increases. Furthermore, it could be seen clearly from the previously mentioned figures that all the distributions display decaying oscillatory behavior.

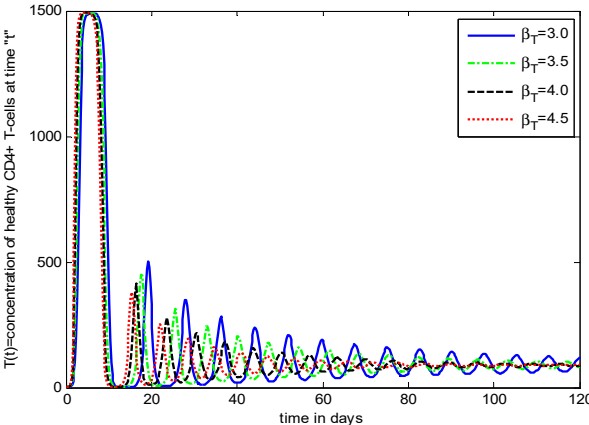

**Figure 1.** The influence of $\beta_T$ on the population dynamics of uninfected cells.

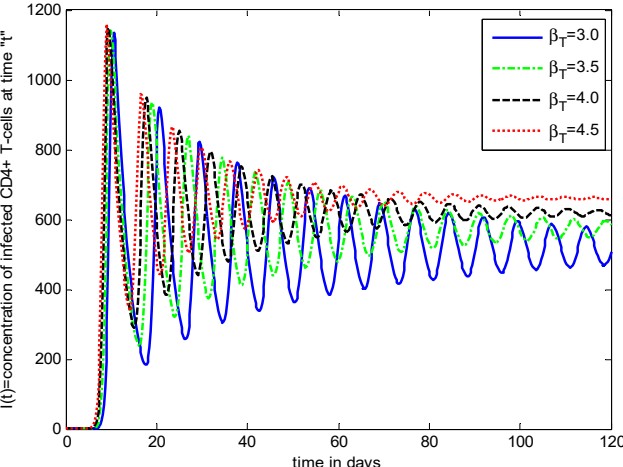

**Figure 2.** The influence of $\beta_T$ on the population dynamics of infected cells.

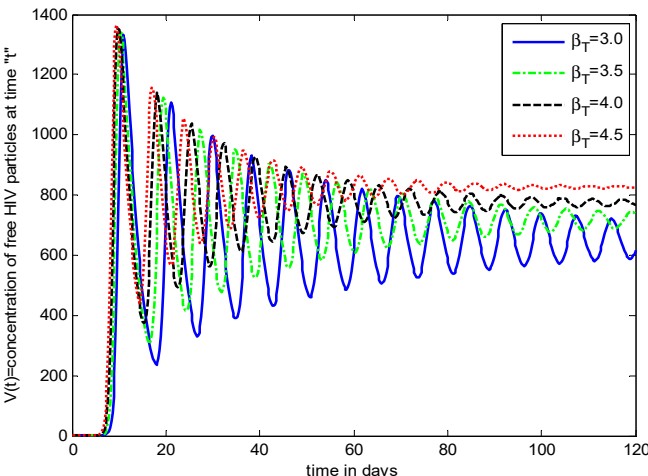

**Figure 3.** The influence of $\beta_T$ on the population dynamics of the HIV virus.

Figures 4–6 show the variation of the death rate of the virus "$\alpha_V$" on the distribution of $T(t)$, $I(t)$, and $V(t)$, respectively. From analysis of the graphs, it is observed that healthy cells and infected cells increased and the virus decreased by increasing the value of "$\alpha_V$". Moreover, notice that the amplitude of oscillations decreases for all cases.

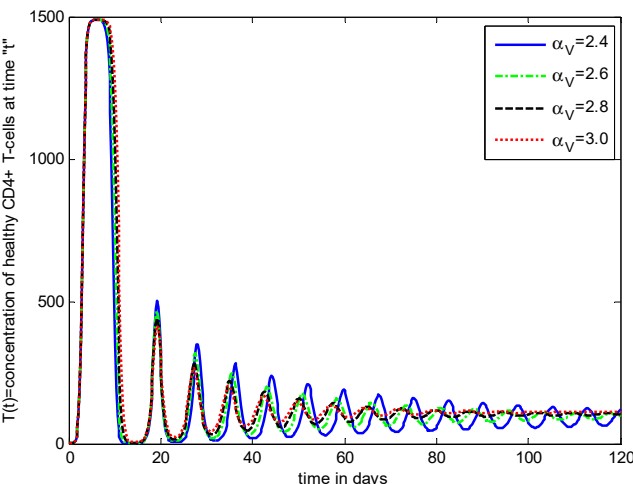

**Figure 4.** The effect of $\alpha_V$ on the density of uninfected cells.

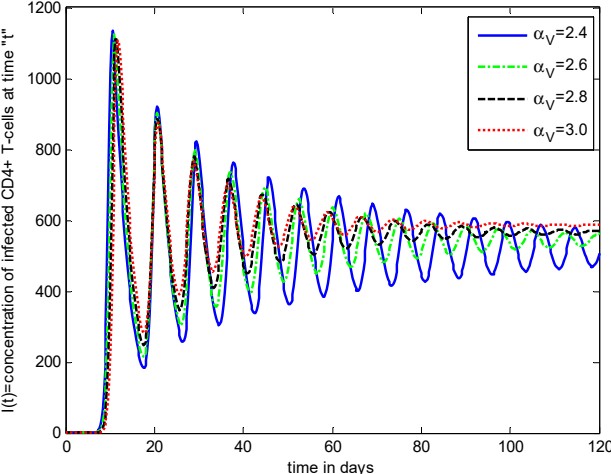

**Figure 5.** The effect of $\alpha_V$ on the density of infected cells.

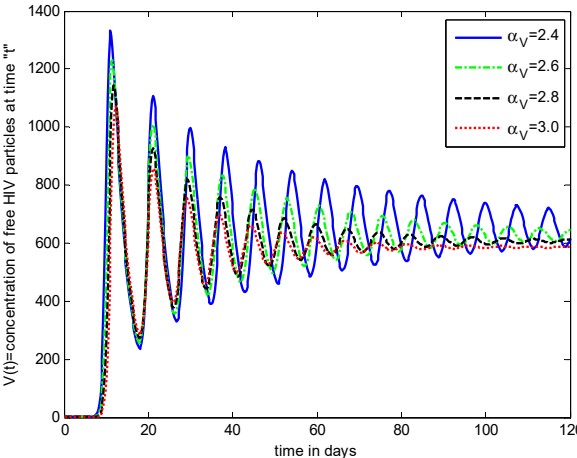

**Figure 6.** The effect of $\alpha_V$ on the density of the HIV virus.

Figures 7–9 represent the influence of "$\phi$" (virus particles released by infected cells) on the population dynamics of healthy cells, infected cells, and the virus. The density of healthy/infected cells and the virus increases by increasing the value of "$\phi$". From the graphs, it is noticed that the healthy cells grow up which improves the resistance against the virus, maintaining the life of the infected patients for some time, but eventually the number of these cells decreases and the disease grows to AIDS which is the crucial condition of the disease.

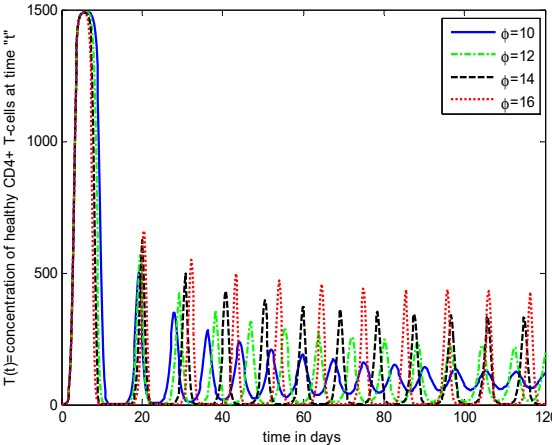

**Figure 7.** Concentration of healthy cells while increasing the value of $\phi$.

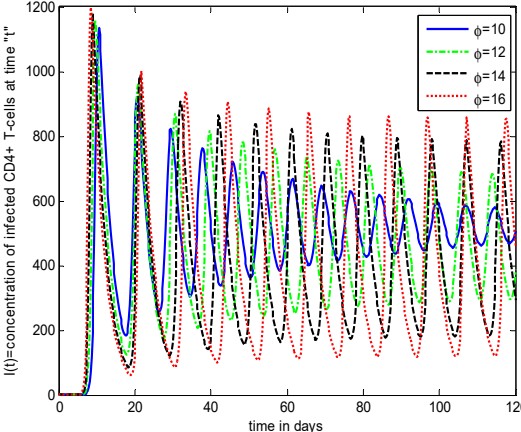

**Figure 8.** Concentration of infected cells while increasing the value of $\phi$.

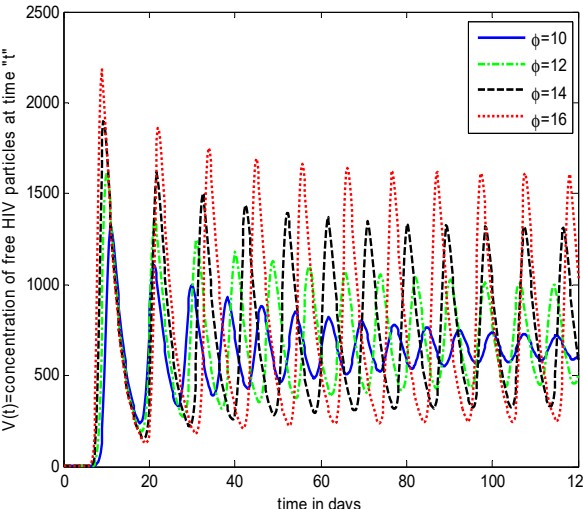

**Figure 9.** The effect of $\phi$ on the density of virus particles.

Figures 10–12 demonstrate the impact of the death rate of infected cells on the dynamic behavior of healthy/infected cells and the virus, respectively. From the figures, it is concluded that when the value of parameter "$r_I$" increases, the strength of healthy/infected cells and the virus increases. Due to an increase in the production of infected cells, the number of healthy cells increases and reaches the highest stage with the passage of time, minimizing the ability of healthy cells to resist against the virus attack, which leaves the body open for disease.

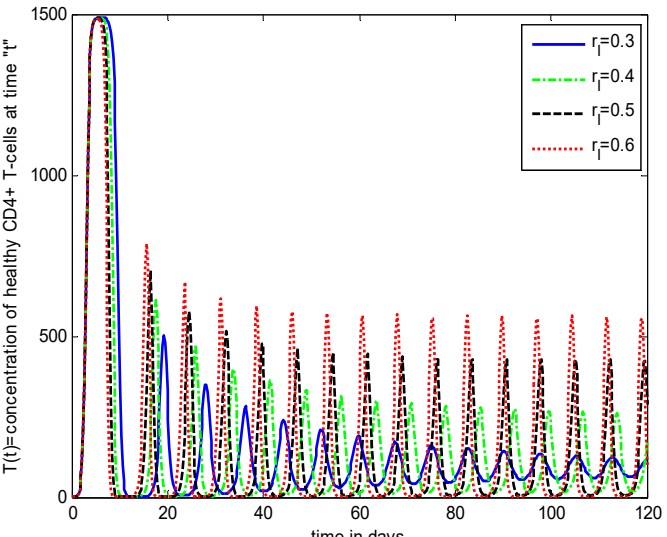

**Figure 10.** The effect of $r_1$ on the density of healthy cells.

Figures 13–15 indicate that when the death rate of healthy cells "$\alpha_T$" increases, the concentration of healthy/infected cells and virus decreases. From the graph, we obtained that the number of healthy cells increases initially, but after some days these cells decrease due to the continuous death of healthy cells and eventually reaches the minimum stage, which shows that the virus attacks the healthy cells. From the aforementioned discussion, all the distributions show decaying oscillatory behaviors that demonstrate the fight among the immune system of the body and the HIV virus. The amplitude of the oscillations gradually decreasing shows that the immunity of the body decreases with the passage of time during infection to fight against the infectious virus. The phase diagram of $I(t)$—$T(t)$, $V(t)$—$T(t)$, $V(t)$—$I(t)$ and $V(t)$—$I(t)$—$T(t)$ are depicted in Figures 16–19, respectively.

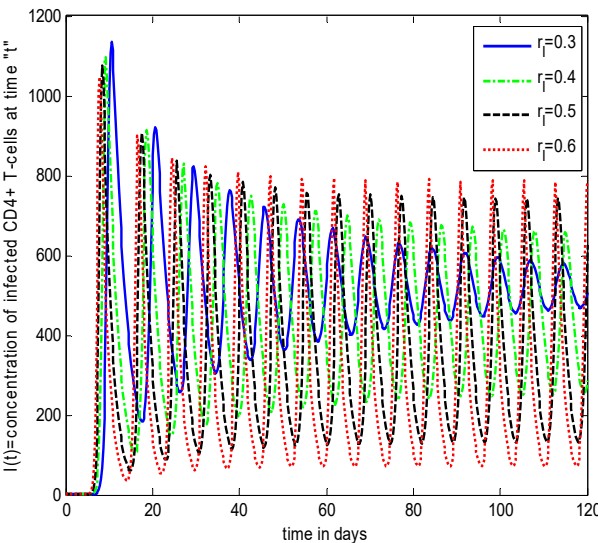

**Figure 11.** The effect of $r_1$ on the density of infected cells.

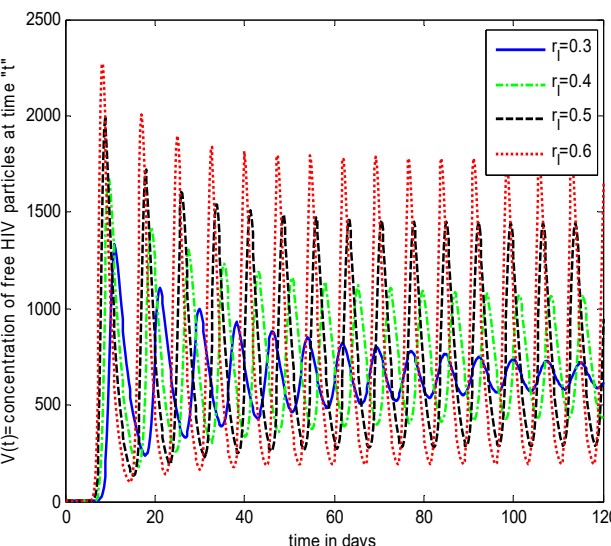

**Figure 12.** The effect of $r_1$ on the density of virus particles.

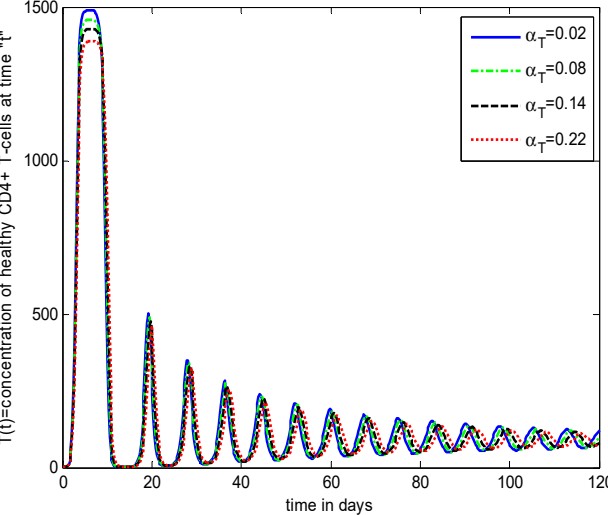

**Figure 13.** The effect of $\alpha_T$ on the density of healthy cells.

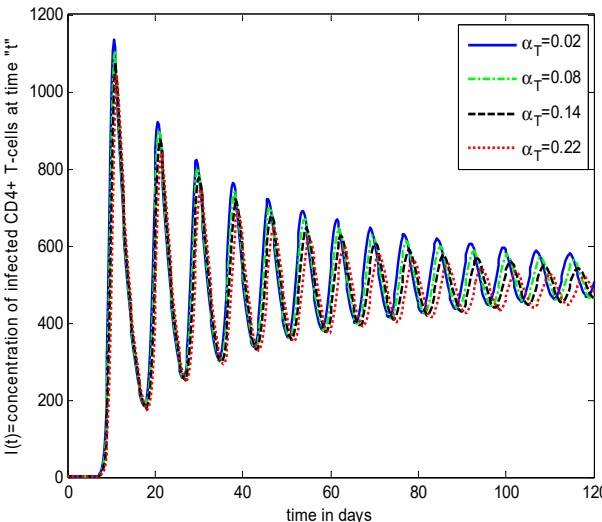

**Figure 14.** The effect of $\alpha_T$ on the density of infected cells.

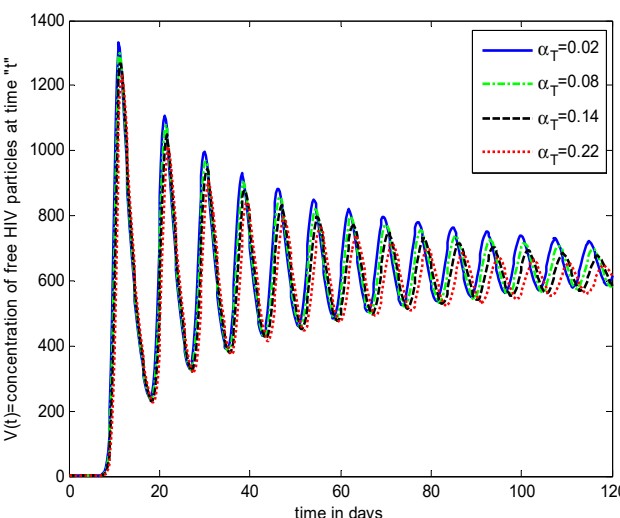

**Figure 15.** The effect of $\alpha_T$ on the density of virus particles.

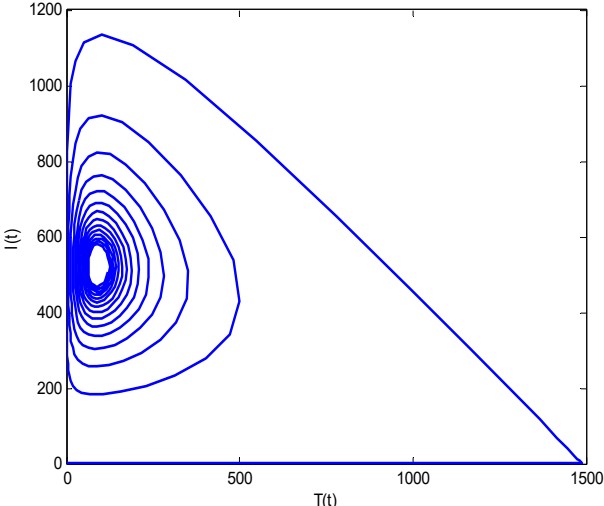

**Figure 16.** The phase diagram $I(t)$ and $T(t)$.

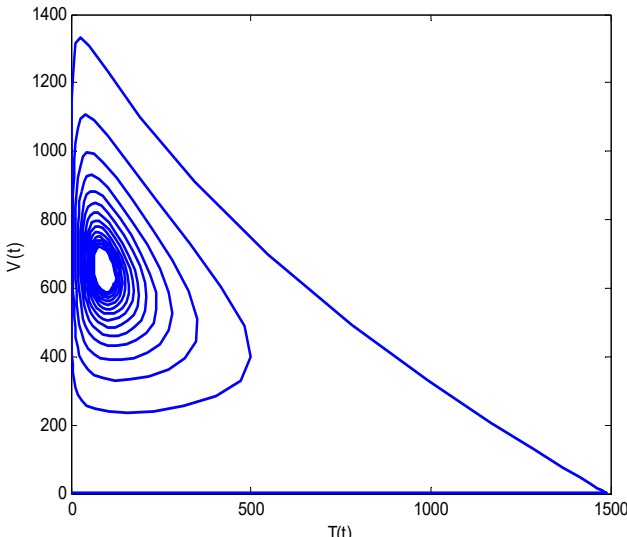

**Figure 17.** The phase diagram of $V(t)$ and $T(t)$.

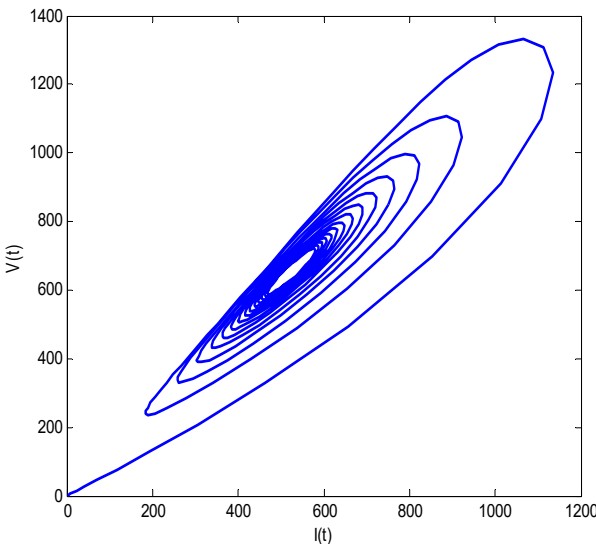

**Figure 18.** The phase diagram of $V(t)$ and $I(t)$.

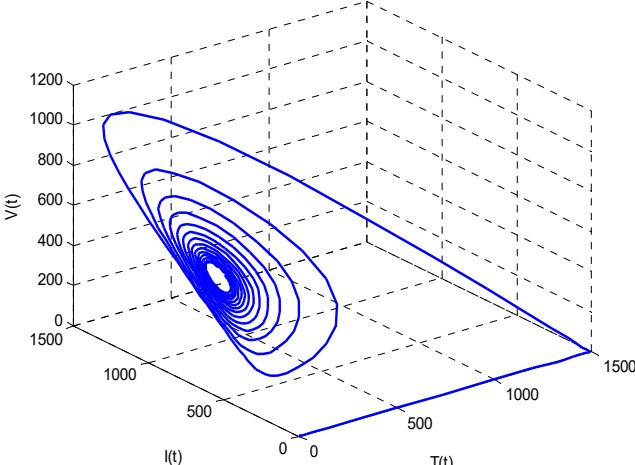

**Figure 19.** The phase diagram of $V(t)$, $I(t)$, and $T(t)$.

## 6. Comparison between the Results of Proposed Method and other Classical Methods

In this section, we describe the comparison of the solutions of the proposed scheme with those solutions already obtained in the literature. In Tables 2–4, the approximate solutions obtained from the proposed scheme are compared with those obtained from the Laplace Adomian decomposition method with Pade approximation [29], the Bessel Collocation method [31], modified variational iteration method [33], an exponential Galerkin method (EGM) [50], and classical Runge–Kutta method of the fourth order. In order to illustrate the solutions, the graphs of the RK4-method solutions and the proposed method solutions are depicted in Figures 20–22, for *T(t)*, *I(t)*, and *V(t)*, respectively. Lastly, for the purpose of having an idea about the accuracy among the approximate solutions obtained from the aforesaid methods relative to the RK4-method, estimations of the absolute errors are demonstrated in Tables 5–7. A close inspection of the results is displayed in Tables 2–7 and Figures 20–22. The results suggest that our approximate solutions are closer to solutions of the RK4-method compared with the solutions of the aforementioned schemes. It is observed that the proposed scheme is reliable for finding approximate solutions to real-world problems.

**Table 2.** Comparison between the results of the cGP(2)-method with other classical methods for $T(t)$.

| $t$ | Runge–Kutta | LADM-Pade [29] | Bessel Coll. N = 8 [30] | PIA(1,1) [31] | MVIM [32] |
|---|---|---|---|---|---|
| 0.2 | 0.2088006789 | 0.2088072731 | 0.2038616561 | 0.2087295073 | 0.2088080868 |
| 0.4 | 0.4062136749 | 0.4061052625 | 0.3803309335 | 0.4059404993 | 0.4062407949 |
| 0.6 | 0.7643508145 | 0.7611467713 | 0.6954623767 | 0.7635790156 | 0.7644287245 |
| 0.8 | 1.4138702489 | 1.3773198590 | 1.2759624442 | 1.4119543417 | 1.4140941730 |
| 1.0 | 2.5911951903 | 2.3291697610 | 2.3832277428 | 2.5867690583 | 2.5919210760 |
| $t$ | DTM N = 6 [33] | EGM N = 3 [50] | EGM N = 4 [50] | EGM N = 5 [50] | cGP(2)-Method |
| 0.2 | 0.2116480000 | 0.2722229510 | 0.2345157340 | 0.1982953765 | 0.2088064964 |
| 0.4 | 0.4226850000 | 0.3065308713 | 0.4201803666 | 0.4183153468 | 0.4062347843 |
| 0.6 | 0.8179400000 | 0.7075440591 | 0.7255920466 | 0.7603331972 | 0.7644082444 |
| 0.8 | 1.5462110000 | 1.5297610198 | 1.4170402360 | 1.4077147917 | 1.4140090611 |
| 1.0 | 2.8540530000 | 2.6678673734 | 2.5916251711 | 2.5915947135 | 2.5915094589 |

**Table 3.** Comparison between the results of the cGP(2)-method with other classical methods for $I(t)$.

| $t$ | Runge–Kutta | LADM-Pade [29] | Bessel coll. N=8 [30] | PIA(1,1) [31] | MVIM [32] |
|---|---|---|---|---|---|
| 0.2 | 0.0000060318 | 0.0000060327 | 0.0000062478 | 0.0000060315 | 0.0000060327 |
| 0.4 | 0.0000131564 | 0.0000131591 | 0.0000129355 | 0.0000131530 | 0.0000131583 |
| 0.6 | 0.0000212206 | 0.0000212683 | 0.0000203526 | 0.0000212101 | 0.0000212233 |
| 0.8 | 0.0000301728 | 0.0000300691 | 0.0000283730 | 0.0000301480 | 0.0000301745 |
| 1.0 | 0.0000400314 | 0.0000398736 | 0.0000369084 | 0.0000399785 | 0.0000400254 |
| $t$ | DTM N = 6 [33] | EGM N = 3 [50] | EGM N = 4 [50] | EGM N = 5 [50] | cGP(2)-Method |
| 0.2 | 0.0000063666 | 0.0000091673 | 0.0000058251 | 0.0000059641 | 0.0000060325 |
| 0.4 | 0.0000139924 | 0.0000155229 | 0.0000134051 | 0.0000131340 | 0.0000131579 |
| 0.6 | 0.0000226514 | 0.0000228459 | 0.0000213405 | 0.0000212682 | 0.0000212231 |
| 0.8 | 0.0000332836 | 0.0000318486 | 0.0000301313 | 0.0000301754 | 0.0000301764 |
| 1.0 | 0.0000485399 | 0.0000421057 | 0.0000400369 | 0.0000400377 | 0.0000400364 |

**Table 4.** Comparison between the results of the cGP(2)-method with other classical methods for $V(t)$.

| $t$ | Runge–Kutta | LADM-Pade [29] | Bessel Coll. N = 8 [30] | PIA(1,1) [31] | MVIM [32] |
|---|---|---|---|---|---|
| 0.2 | 0.0618808474 | 0.0618799602 | 0.0618799185 | 0.0618796999 | 0.0618799087 |
| 0.4 | 0.0382961304 | 0.0383132488 | 0.0382949349 | 0.0382939096 | 0.0382959576 |
| 0.6 | 0.0237057031 | 0.0243917434 | 0.0237043186 | 0.0237016917 | 0.0237102948 |
| 0.8 | 0.0146813143 | 0.0099672189 | 0.0146795698 | 0.0146744145 | 0.0147004190 |
| 1.0 | 0.0091015791 | 0.0033050764 | 0.0090993030 | 0.0090905052 | 0.0091572387 |
| $t$ | DTM N = 6 [33] | EGM N = 3 [50] | EGM N = 4 [50] | EGM N = 5 [50] | cGP(2)-Method |
| 0.2 | 0.0618800000 | 0.0618823466 | 0.0618790041 | 0.0618799035 | 0.0618799805 |
| 0.4 | 0.0383090000 | 0.0383077329 | 0.0382950148 | 0.0382947890 | 0.0382950575 |
| 0.6 | 0.0239200000 | 0.0237055266 | 0.0237053683 | 0.0237046061 | 0.0237047074 |
| 0.8 | 0.0162120000 | 0.0146708169 | 0.0146798882 | 0.0146803810 | 0.0146804932 |
| 1.0 | 0.0160500000 | 0.0091056907 | 0.0091009339 | 0.0091008486 | 0.0091009447 |

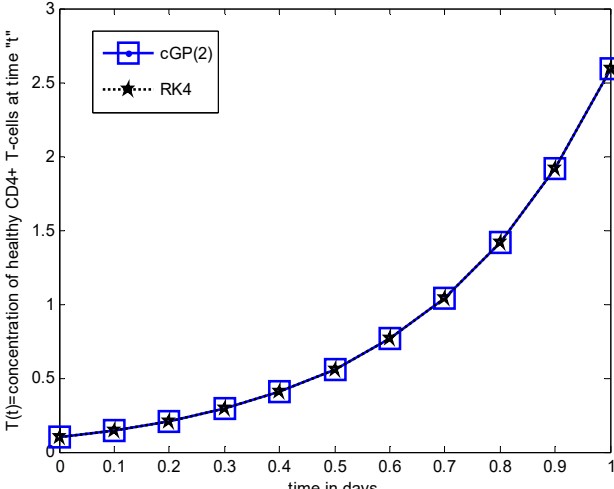

**Figure 20.** Comparative analysis of Galerkin and RK4 schemes from $T(t)$.

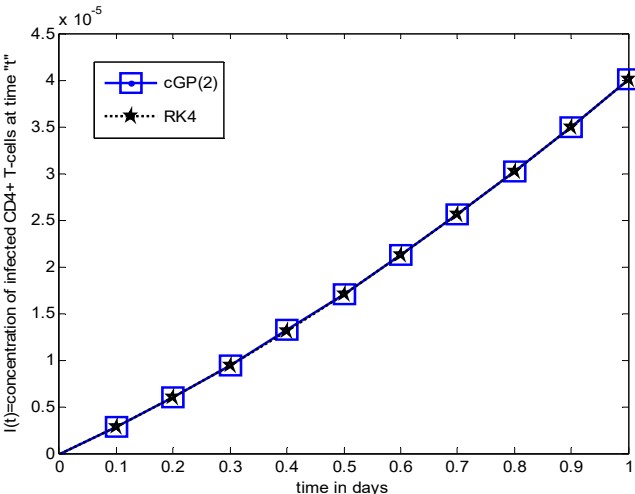

**Figure 21.** Comparative analysis of Galerkin and RK4 schemes from $I(t)$.

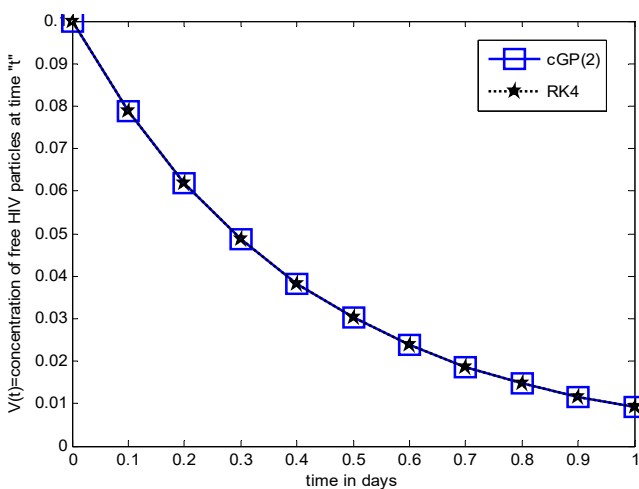

**Figure 22.** Comparative analysis of Galerkin and RK4 schemes for $V(t)$.

**Table 5.** Comparison of absolute errors for $T(t)$ of the cGP(2) and classical methods relative to the RK4-method.

| $t$ | LADM-Pade [29] | Bessel Coll. N = 8 [30] | PIA(1,1) [31] | MVIM [32] | DTM N = 6 [33] |
|---|---|---|---|---|---|
| 0.2 | 0.000006594223280 | 0.004939022776720 | 0.000071171576720 | $7.40792327999 \times 10^{-6}$ | 0.002847321123280 |
| 0.4 | 0.000108412464641 | 0.025882741464641 | 0.000273175664641 | $2.71199353589 \times 10^{-5}$ | 0.016471325035359 |
| 0.6 | 0.003204043237096 | 0.068888437837096 | 0.000771798937096 | $7.79099629040 \times 10^{-5}$ | 0.053589185462904 |
| 0.8 | 0.036550389916353 | 0.137907804716353 | 0.001915907216353 | $2.23924083647 \times 10^{-3}$ | 0.132340751083647 |
| 1.0 | 0.262025429366243 | 0.207967447566243 | 0.004426132066243 | $7.25885633757 \times 10^{-3}$ | 0.262857809633757 |
| $t$ | EGM N = 3 [50] | EGM N = 4 [50] | EGM N = 5 [50] | cGP(2)-Method | |
| 0.2 | 0.063422272123280 | 0.025715055123280 | 0.010505302376720 | $5.81760745974 \times 10^{-6}$ | |
| 0.4 | 0.099682803664641 | 0.013966691635359 | 0.012101671835359 | $2.11093478030 \times 10^{-5}$ | |
| 0.6 | 0.056806755437096 | 0.038758767937096 | 0.004017617337096 | $5.74299020841 \times 10^{-5}$ | |
| 0.8 | 0.115890770883647 | 0.003169987083647 | 0.006155457216353 | $1.38812210279 \times 10^{-4}$ | |
| 1.0 | 0.076672183033757 | 0.000429980733757 | 0.000399523133757 | $3.14268571135 \times 10^{-4}$ | |

**Table 6.** Comparison of absolute errors for $I(t)$ of the cGP(2) and classical methods relative to the RK4-method.

| $t$ | LADM-Pade [29] | Bessel Coll. N = 8 [30] | PIA(1,1) [31] | MVIM [32] | DTM N= 6 [33] |
|---|---|---|---|---|---|
| 0.2 | $8.21028439 \times 10^{-10}$ | $2.15921028439 \times 10^{-7}$ | $3.7897156100 \times 10^{-10}$ | $8.2102843899 \times 10^{-10}$ | $3.3472103 \times 10^{-7}$ |
| 0.4 | $2.61393801 \times 10^{-9}$ | $2.20986061988 \times 10^{-7}$ | $3.4860619888 \times 10^{-9}$ | $1.8139380112 \times 10^{-9}$ | $8.3591393 \times 10^{-7}$ |
| 0.6 | $4.76223967 \times 10^{-8}$ | $8.68077603328 \times 10^{-7}$ | $1.0577603329 \times 10^{-8}$ | $2.6223966711 \times 10^{-9}$ | $1.4307224 \times 10^{-6}$ |
| 0.8 | $1.03710297 \times 10^{-7}$ | $1.79981029736 \times 10^{-6}$ | $2.4810297362 \times 10^{-8}$ | $1.6897026384 \times 10^{-9}$ | $3.1107898 \times 10^{-6}$ |
| 1.0 | $1.57815845 \times 10^{-7}$ | $3.12301584505 \times 10^{-6}$ | $5.2915845057 \times 10^{-8}$ | $6.0158450573 \times 10^{-9}$ | $8.5084842 \times 10^{-6}$ |
| $t$ | EGM N = 3 [50] | EGM N = 4 [50] | EGM N = 5 [50] | cGP(2)-Method | |
| 0.2 | $3.1354210284 \times 10^{-6}$ | $2.0677897156 \times 10^{-7}$ | $6.77789715609 \times 10^{-8}$ | $6.6752196826462 \times 10^{-10}$ | |
| 0.4 | $2.3664139380 \times 10^{-6}$ | $2.4861393801 \times 10^{-7}$ | $2.24860619888 \times 10^{-8}$ | $1.4924805624754 \times 10^{-9}$ | |
| 0.6 | $1.6252223967 \times 10^{-6}$ | $1.1982239667 \times 10^{-7}$ | $4.75223966710 \times 10^{-8}$ | $2.4821350652576 \times 10^{-9}$ | |
| 0.8 | $0.1.67578970 \times 10^{-6}$ | $4.1510297361 \times 10^{-8}$ | $2.58970263849 \times 10^{-9}$ | $3.6558718785258 \times 10^{-9}$ | |
| 1.0 | $2.0742841549 \times 10^{-6}$ | $5.4841549427 \times 10^{-9}$ | $6.28415494269 \times 10^{-9}$ | $5.0422736406875 \times 10^{-9}$ | |

**Table 7.** Comparison of absolute errors for $V(t)$ of the cGP(2) and classical methods relative to the RK4-method.

| $t$ | LADM-Pade [29] | Bessel Coll. N = 8 [30] | PIA(1,1) [31] | MVIM [32] | DTM N = 6 [33] |
|---|---|---|---|---|---|
| 0.2 | $8.87201671 \times 10^{-7}$ | $9.28901670999 \times 10^{-7}$ | $1.14750167099 \times 10^{-6}$ | $9.3870167099 \times 10^{-7}$ | $8.47401671 \times 10^{-7}$ |
| 0.4 | $1.71183628 \times 10^{-5}$ | $1.19553719699 \times 10^{-6}$ | $2.22083719699 \times 10^{-6}$ | $1.7283719699 \times 10^{-7}$ | $1.28695628 \times 10^{-5}$ |
| 0.6 | $6.86040272 \times 10^{-4}$ | $1.38452847400 \times 10^{-6}$ | $4.01142847400 \times 10^{-6}$ | $4.5916715259 \times 10^{-6}$ | $2.14296871 \times 10^{-4}$ |
| 0.8 | $4.71409543 \times 10^{-3}$ | $1.74452298399 \times 10^{-6}$ | $6.89982298399 \times 10^{-6}$ | $1.9104677016 \times 10^{-5}$ | $1.53068567 \times 10^{-3}$ |
| 1.0 | $5.79650268 \times 10^{-3}$ | $2.27607680900 \times 10^{-6}$ | $1.10738768090 \times 10^{-5}$ | $5.5659623191 \times 10^{-5}$ | $6.94842092 \times 10^{-3}$ |

| $t$ | EGM N = 3 [50] | EGM N = 4 [50] | EGM N = 5 [50] | cGP(2)-Method |
|---|---|---|---|---|
| 0.2 | $1.499198329 \times 10^{-6}$ | $1.8433016709 \times 10^{-6}$ | $9.43901670998 \times 10^{-7}$ | $8.6688500106069 \times 10^{-7}$ |
| 0.4 | $1.160246280 \times 10^{-5}$ | $1.1156371969 \times 10^{-6}$ | $1.34143719699 \times 10^{-6}$ | $1.0728542867849 \times 10^{-7}$ |
| 0.6 | $1.765284740 \times 10^{-7}$ | $3.3482847399 \times 10^{-7}$ | $1.09702847399 \times 10^{-6}$ | $9.9566078337957 \times 10^{-7}$ |
| 0.8 | $1.049742298 \times 10^{-5}$ | $1.4261229840 \times 10^{-6}$ | $9.33322984000 \times 10^{-7}$ | $8.2106806348695 \times 10^{-7}$ |
| 1.0 | $4.111623191 \times 10^{-6}$ | $6.4517680900 \times 10^{-7}$ | $7.30476809001 \times 10^{-7}$ | $6.3432370331871 \times 10^{-7}$ |

## 7. Conclusions

The aim of the present study was to modify the HIV model and introduce a variable source term for the generation of healthy cells dependent on the virus and numerically approximate the density of healthy cells, infected cells, and free HIV virus after infection. We implemented a new numerical technique, namely, the continuous Galerkin–Petrov scheme for efficient and accurate solutions with the aforementioned model In addition, the impact of various physical parameters was analyzed, and all the observations were presented graphically. The main findings of the current study are summarized as follows:

i.    Increasing growth rate of healthy cells, ($\beta_T$), shows a decreasing effect in the population dynamics of healthy cells, while showing an increasing effect in the population dynamics of infected cells and HIV particles. All the profiles showed a decaying oscillatory behavior.

ii.   The healthy cells and infected cells show an increasing effect, while free virus distribution shows a decreasing behavior with an increase in the values of the virus death rate ($\alpha_V$).

iii.  It is noticed that the virus particles released by infected cells ($\phi$) show significant variations in the population distributions of healthy cells, infected cells, and the virus. By increasing the value of "$\phi$", the healthy cells, infected cells, and the virus increases.

iv.   The graphical trends illustrate increased decay in distributions of all dependent variables with an increase in the death rate of infected cells ($r_I$).

v.    The decrease in the density of healthy cells, infected cells, and free HIV particles is observed by increasing "$\alpha_T$".

Moreover, we performed an analysis for the reproduction of number "$R_0$" and concluded that when "$R_0 \leq 1$" the disease-free equilibrium state "$E_0$" is locally as well as globally stable, while when "$R_0 > 1$", then "$E_0$" is locally and globally unstable.

Furthermore, the well-known classical method for the initial value problem, namely, the RK-method, is utilized for the model and all the outcomes were compared with those from the proposed scheme and other schemes available in the open literature. It was observed that the suggested scheme is efficient and accurate in comparison with the other schemes. The proposed scheme could be applicable for solving complex real-world problems. In the future, the presented work is extendable in different directions. For instance, the authors are interested to determine the most proper and effective method of vaccination and treatment using the model.

**Author Contributions:** Conceptualization, A., S.A., M.F.Y. and D.P.; Formal analysis, A., S.A., M.F.Y. and D.P.; Investigation, S.A.; Methodology, A., Z., M.T.K., S.A. and D.P.; Resources, A. and M.F.Y.; Software, A., Z. and M.F.Y.; Supervision, M.F.Y.; Validation, A. and M.T.K.; Writing—original draft, A. and M.T.K.; Writing—review & editing, A. and D.P. All authors have read and agreed to the published version of the manuscript.

**Funding:** This research was supported by the Fundamental Fund of Khon Kaen University. This research has received funding support from the National Science, Research and Innovation Fund or NSRF.

**Institutional Review Board Statement:** Not Applicable.

**Informed Consent Statement:** Not Applicable.

**Data Availability Statement:** Not Applicable.

**Conflicts of Interest:** The authors declare no conflict of interest.

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
