# Peer review of "A Computational Approach to a Model for HIV and the Immune System Interaction"

_axioms, doi:10.3390/axioms11100578_

Round 1

Reviewer 1 Report

I have read and reviewed the paper “A Computational Approach to a Model for HIV and the Immune System Interaction” very carefully and found that the results of the paper are looking interesting and mathematically correct. The novelty of the results is also good and the presentation of the paper is suitable. But for the final publication, the present form of the paper required some revisions. My suggestions are given as follows:

1.      Please check the manuscript carefully for typos and grammar errors.

2.      The abstract should be concise and informative.

3.      Complete the introduction by including the latest published works in the literature regarding this area of study.

4.      I suggest you provide more details and comparisons in the introduction for how this paper is distinguished from other works in the literature.

5.  The titles should be changed

6.      At the end of all the equations, the commas or dots are missing as per typing rule, so check and correct.

7.  The similarity rate is %36. Please reduce it.

8.      The advantages of considering the proposed numerical scheme over the existing methods should be included in the conclusion part.

The results are correct. I recommend it for publication after the above-suggested revisions.

Author Response

Manuscript ID: axioms-1936965

A Computational Approach to a Model for HIV and the Immune System Interaction

Response to Reviewers

First of all, we would like to thank all the anonymous reviewers and the Editor-in-Chief for thorough reading of this manuscript and for the thoughtful comments and constructive suggestions, which help to improve the quality of this manuscript. We considered all the suggestions positive and tried our best to show these changes in our manuscript. Our are response follows:

Response to Editor-in-Chief

Dear Professor,

Thank you for giving us the opportunity to submit a revised draft of our manuscript titled “A Computational Approach to a Model for HIV and the Immune System Interaction” to Axioms. We appreciate the time and efforts that you and the reviewers have dedicated for providing your valuable feedback on our manuscript. We are grateful to the reviewers for their insightful comments on our paper. We have been able to incorporate changes to reflect most of the suggestions provided by the reviewers. Here is a point-by-point response to the reviewers’ comments and concerns.

Response to Reviewer 1

I have read and reviewed the paper “A Computational Approach to a Model for HIV and the Immune System Interaction” very carefully and found that the results of the paper are looking interesting and mathematically correct. The novelty of the results is also good and the presentation of the paper is suitable. But for the final publication, the present form of the paper required some revisions. My suggestions are given as follows:

Response: Thank you very much for your positive and insightful comment. The recommendations are valuable and will help us in our future work. Some of the excellent guidelines you have provided are quite beneficial to our work. The points suggested in various parts of the article are extremely significant in strengthening the quality of the manuscript. We tried our best to include all of these ideas into the revised version. Hopefully, it is now in satisfactory quality:

Comment 1. Please check the manuscript carefully for typos and grammar errors.

Response to 1: Thanks for your constructive comment!

The issue highlighted is extremely useful in improving the article's quality. As suggested by the learned reviewer, the typos and language errors have been removed. The quality of the language has been checked throughout the paper. The grammar of language has been corrected. Similarly, the typos mistakes have been removed to make the paper useful for the readers.

Comment 2. The abstract should be concise and informative.

Response to 2: Thank you for your positive comment!

The point raised is very helpful in taking the article to better standards. We incorporated the changes suggested by the learned reviewer in the revised version of the manuscript.

Comment 3.  Complete the introduction by including the latest published works in the literature regarding this area of study.

Response to 3: Thank you for your positive comment!

Thank you very much for acknowledging our research and for your positive & valuable suggestions. The suggestions are worthy and have an important guiding significance to our work. We completed the introduction by including the latest published works.

Comment 4. I suggest you provide more details and comparisons in the introduction for how this paper is distinguished from other works in the literature.

Response to 4: Thank you for your positive comment!

We incorporated the comparative analysis in the revised manuscript. We appreciate your for your positive & valuable suggestions. The suggestions are worthy and have an important guiding significance. We did our best to accommodate the suggestion in the revised version. Hopefully, now it is in better shape.

Comment 5. The titles should be changed

Response 5: Thanks for your constructive comment!

It is for your kind information that the paper is extracted from thesis of the student (Muhammd Tufail Khan, Department of Mathematics & Statistics, Bacha Khan University, Charsadda 24461, Pakistan). Therefore, we cannot change the title.

Comment 6. At the end of all the equations, the commas or dots are missing as per typing rule, so check and correct.

Response to 6: Thanks for the thoughtful comment!

All the equations has been checked throughout the paper. The commas and dot after the equations has been corrected. Similarly, the typos mistakes have been removed to make the paper useful for the readers.

Comment 7. The similarity rate is %36. Please reduce it.

Response to 7: Thank you for your guidance!

We revised the manuscript and now the similarity is reduced. The paper is extracted from the thesis of student and submitted to Bacha Khan University Charsdada.

Comment 8. The advantages of considering the proposed numerical scheme over the existing methods should be included in the conclusion part.

Response to 8: Thank you very much for your positive & valuable comments. The points raised are very helpful in taking our research to better standards. We incorporated suggested changes in the conclusion accordingly. Hopefully, the revised structure of conclusion is clear and transparent.

Comment 9.  The results are correct. I recommend it for publication after the above-suggested revisions.

Response to 9: Thank you for your very precious idea and highly acknowledge its worth. Thanks for your valuable comments and suggestions for the improvement of our paper. We corrected the mentioned above comments accordingly in the revised manuscript.

We are again very thankful to the reviewers and the editor for taking an interest in improving our manuscript and for suggestions and enhancing the manuscript. All the changes have been made in the article as directed. We hope that the article is now well in accordance.

Reviewer 2 Report

In the work "A Computational Approach to a Model for HIV and the Immune System Interaction" the numerical solution of the Human Immunodeficiency Virus (HIV) infection model which is major problem for global public healthy. Acquired Immunodeficiency Syndrome (AIDS) is a communicable disease and HIV is the causative agent for AIDS which damages ability of body to fights against disease and easily usual innocuous infections attack on the body. On entering the body, HIV infects a large amount of CD4+ T-cells and disturbs the supply rate of these cells from thymus. Herein, we consider the model with variable source term in which the production of these cells as a monotonically decreasing function of viral load I recommend the following changes

1. The abstract is too long.

2. Add main contribution in the abstract only.

3. Add the motivation and gap in the introduction part.

4. The considered system is a system of ODEs. We can solve the the system by RK-4 method but Galerkin's method is also used why? Comments.

5. The following refs are related to the work, it is advised to add in the revised manuscript

(a) Local radial basis function-finite difference based algorithms forsingularly perturbed Burgers’ model, Mathematics and Computers in Simulation, 198, (2022) 106-126

(b) A cubic B-spline quasi-interpolation algorithm to capture the pattern formation of coupled reaction diffusion models, Engineering with Computers, (2021) doi.org/10.1007/s00366-020-01278-3.

(c) Local radial basis functions and scale-3 Haar wavelets operational matrices based numerical algorithms for generalized regularized long wave model, Wave Motion, 109 (2022) 102846.

(d) New Scale-3 Haar Wavelets Algorithm for Numerical Simulation of Second Order Ordinary Differential Equations, Proceedings of the National Academy of Sciences, India Section A: Physical Sciences, (2018) doi.org/10.1007/s40010- 018-0538-y. (

Author Response

Manuscript ID: axioms-1936965

A Computational Approach to a Model for HIV and the Immune System Interaction

Response to Reviewers

First of all, we would like to thank all the anonymous reviewers and the Editor-in-Chief for thorough reading of this manuscript and for the thoughtful comments and constructive suggestions, which help to improve the quality of this manuscript. We considered all the suggestions positive and tried our best to show these changes in our manuscript. Our are response follows:

Response to Editor-in-Chief

Dear Professor,

Thank you for giving us the opportunity to submit a revised draft of our manuscript titled “A Computational Approach to a Model for HIV and the Immune System Interaction” to Axioms. We appreciate the time and efforts that you and the reviewers have dedicated for providing your valuable feedback on our manuscript. We are grateful to the reviewers for their insightful comments on our paper. We have been able to incorporate changes to reflect most of the suggestions provided by the reviewers. Here is a point-by-point response to the reviewers’ comments and concerns.

Response to Reviewer 2

In the work "A Computational Approach to a Model for HIV and the Immune System Interaction" the numerical solution of the Human Immunodeficiency Virus (HIV) infection model which is major problem for global public healthy. Acquired Immunodeficiency Syndrome (AIDS) is a communicable disease and HIV is the causative agent for AIDS which damages ability of body to fights against disease and easily usual innocuous infections attack on the body. On entering the body, HIV infects a large amount of CD4+ T-cells and disturbs the supply rate of these cells from thymus. Herein, we consider the model with variable source term in which the production of these cells as a monotonically decreasing function of viral load I recommend the following changes

Response: Thank you very much for appreciating our research and for your positive and insightful ideas. The recommendations are valuable and will help us in our future work. Some of the excellent guidelines you have provided are quite beneficial to our work. The points suggested in various parts of the article are extremely significant in strengthening the quality of the manuscript. We tried our best to include all of these ideas into the revised version. Hopefully, it is now in satisfactory quality:

Comment 1. The abstract is too long.

Response to 1: Thanks for your constructive comment!

As suggested by the learned reviewer, the abstract is now retained brief and straightforward. Hopefully, now is well accordance.

Comment 2. Add main contribution in the abstract only.

Response to 2: Thanks for the thoughtful comment!

We incorporated the suggested changes in the revised version of the manuscript. We appreciate your constructive and beneficial suggestions.

Comment 3. Add the motivation and gap in the introduction part.

Response to 3: Thank you for your positive comment!

The point raised is very helpful in taking the article to better standards. We improved the motivation of the manuscript suggested by learned reviewer. Hopefully, now it is in better shape.

Comment 4. The considered system is a system of ODEs. We can solve the the system by RK-4 method but Galerkin's method is also used why? Comments.

Response to 4: Thank you for your positive comment!

We implemented the continuous Galerkin-Petrov scheme of polynomial order two simply cGP(2) scheme for finding the approximate solution of the suggested non linear model. Therefore, to having an idea about the accuracy and reliability of the approximate solutions obtained from the proposed technique, the well-known classical RK4-method is also implemented. Estimated the absolute errors between the results of both schemes to better understand and authenticate our proposed method's applicability in practical problems. We examined the accuracy of the cGP(2) technique and observed that it produces more accurate results at relatively longer step sizes than the RK-4 scheme. From the numerical and graphical comparison of the results of both schemes, it could be notice-worthy that the results are in good agreement with each other. The proposed scheme could be applicable for solving complex real-world problems.

Comment 5. The following refs are related to the work, it is advised to add in the revised manuscript.

(a) Local radial basis function-finite difference based algorithms forsingularly perturbed Burgers’ model, Mathematics and Computers in Simulation, 198, (2022) 106-126

(b) A cubic B-spline quasi-interpolation algorithm to capture the pattern formation of coupled reaction diffusion models, Engineering with Computers, (2021) doi.org/10.1007/s00366-020-01278-3.

(c) Local radial basis functions and scale-3 Haar wavelets operational matrices based numerical algorithms for generalized regularized long wave model, Wave Motion, 109 (2022) 102846.

(d) New Scale-3 Haar Wavelets Algorithm for Numerical Simulation of Second Order Ordinary Differential Equations, Proceedings of the National Academy of Sciences, India Section A: Physical Sciences, (2018) doi.org/10.1007/s40010- 018-0538-y. (

Response to 5: We included the mentioned above articles. Once again thank you for your valuable comments and suggestions for the improvement of our paper. It will enrich the introduction of the current research article.

We are again very thankful to the reviewers and the editor for taking an interest in improving our manuscript and for suggestions and enhancing the manuscript. All the changes have been made in the article as directed. We hope that the article is now well in accordance.

Reviewer 3 Report

The manuscript is not well prepared. The Math equations is hard to read, especially when involving subscript and superscript. The caption of figures got many mistakes.

1.       Many text in introduction are copy and paste from

Attaullah, R. D., & Weera, W. (2022). Galerkin time discretization scheme for the transmission dynamics of HIV infection with non-linear supply rate. Journal: AIMS Mathematics, (6), 11292-11310.

This is not allowed. Please rewrite the introduction.

2.       The contribution of this manuscript is not clear.  The math presented here is very common and well established.

3.       Please make sure the superscript and subscript had been used properly, such as CD4+ and R0, the + and 0 should be superscript and subscript respectively.

4.       Page 6 Table 1. Please cite where you get this Table 1 ., especially from where you get those values.

5.       Section 6. It is a bit strange that the whole section is explaining those Figure in Section 7. Please if possible, name the figures as 6.X.  But the caption of figures are 1,2,3… . This did not match. Please make correction.

6.       Page 15 : The caption and the graph are not match for Fig 9, 10, 11,12. Please correct it. Also Check Fig 16-22. I guess all the caption are not correct.

7.       Page 18, Table 7.4.  If possible, please write the absolute errors using standard form, as those you used in Table 7.5.

8.       No need separate section for 9. Future Recommendations, because only two lines, just combine it with conclusion.

9.       Reference : Ref 50 do not have enough info such as volume, issue, page.  Furthermore, I unable to find the Ref 26, 27, 28 in the text. Please check.

Author Response

Manuscript ID: axioms-1936965

A Computational Approach to a Model for HIV and the Immune System Interaction

Response to Reviewers

First of all, we would like to thank all the anonymous reviewers and the Editor-in-Chief for thorough reading of this manuscript and for the thoughtful comments and constructive suggestions, which help to improve the quality of this manuscript. We considered all the suggestions positive and tried our best to show these changes in our manuscript. Our are response follows:

Response to Editor-in-Chief

Dear Professor,

Thank you for giving us the opportunity to submit a revised draft of our manuscript titled “A Computational Approach to a Model for HIV and the Immune System Interaction” to Axioms. We appreciate the time and efforts that you and the reviewers have dedicated for providing your valuable feedback on our manuscript. We are grateful to the reviewers for their insightful comments on our paper. We have been able to incorporate changes to reflect most of the suggestions provided by the reviewers. Here is a point-by-point response to the reviewers’ comments and concerns.

Response to Reviewer 3

The manuscript is not well prepared. The Math equations is hard to read, especially when involving subscript and superscript. The caption of figures got many mistakes.

Response: Thank you very much for your insightful and constructive comments and suggestions. The suggestions are worthy and have an important guiding significance to our work. We corrected the subscript and superscript in the whole manuscript. Some helpful directions you've pointed out are of much value to our work. The points raised in different sections of the article are very helpful in taking the report to better standards. We did our best to accommodate all these suggestions in the revised version. Hopefully, now it is in qualitative shape:

Comment 1. Many text in introduction are copy and paste from

Attaullah, R. D., & Weera, W. (2022). Galerkin time discretization scheme for the transmission dynamics of HIV infection with non-linear supply rate. Journal: AIMS Mathematics, (6), 11292-11310.

This is not allowed. Please rewrite the introduction.

Response to 1: Thanks for the thoughtful comment!

The aforementioned points raised in the article are essential and excellent. It will enrich the current research article. We checked the whole manuscript for similar sentences and rewrite as suggested by the learned reviewer.  Hopefully, now it is in better shape.

Comment 2. The contribution of this manuscript is not clear.  The math presented here is very common and well established.

Response to 2:  Thank you very much for the essential and excellent question. In the existing literature, most of the authors assumed the dynamics of HIV with the supply rate of new CD4+T-cells from the thymus as constant. However, the infection of HIV having the potential to infect these cells, and variable phenomena have been observed instead of constant Therefore, we introduced a novel idea of nonlinear variable source term for the generation of new CD4+T-cells from thymus in the dynamics of HIV infection to represent more realistic phenomena.

Implemented a new method, namely the continuous Galerkin-Petrov scheme, for finding the approximate solution of the afore-mentioned nonlinear model. Studied the impact of different parameters involved in the model on the population dynamics of healthy T-cells, infected T-cells, and free viruses. Medically, it implies that some parameter values can cause the cell and virus population to fluctuate. Moreover, provide sufficient information to clinicians to reduce the viral load of the infection.

The model, as mentioned earlier, is vital in the field of mathematical modeling of HIV infection of CD4+T-cells. This will be used to analyze the population dynamics of CD4+T-cells in the presence and absence of HIV, helpful to observe the symptoms of AIDS seen clinically and valuable to hold back the disease. This study will be a valuable addition to the current literature on biomathematics.

Comment 3. Please make sure the superscript and subscript had been used properly, such as CD4+ and R0, the + and 0 should be superscript and subscript respectively.

Response to 3: Thank you for your guidance!

The changes have been made as suggested by the learner reviewer. Once again thank you for your valuable comments and suggestions for the improvement of our paper. We have done the correction accordingly, and hopefully, now it is well in accordance.

Comment 4. Page 6 Table 1. Please cite where you get this Table 1, especially from where you get those values.

Response to 4: Thank you very much for your positive & valuable comments. The points raised are very helpful in taking our research to better standards. We cited the values of the Table-1 in the revised manuscript. Hopefully, the revised structure is clear and transparent.

Comment 5.    Section 6. It is a bit strange that the whole section is explaining those Figure in Section 7. Please if possible, name the figures as 6.X.  But the caption of figures are 1,2,3… . This did not match. Please make correction.

Response to 5: Thank you for your very precious idea and highly acknowledge its worth. We corrected accordingly in the revised manuscript.

Comment 6.  Page 15 : The caption and the graph are not match for Fig 9, 10, 11,12. Please correct it. Also Check Fig 16-22. I guess all the caption are not correct. 

Response to 6: Thanks for your insightful comments. We corrected all the captions of the figures accordingly in the revised version. Now all the caption and figures are matching.

Comment 7.  Page 18, Table 7.4.  If possible, please write the absolute errors using standard form, as those you used in Table 7.5.

Response to 7: Thanks for the comments. It is for your kind information that in Tables 7.2--7.4 (currently Table 2--4), we compared the results obtained through different methods whereas in Table 7.5 (currently Table 5), we compared the absolute errors relative to RK-method.

Comment 8.  No need separate section for 9. Future Recommendations, because only two lines, just combine it with conclusion.

Response to 8: Thanks for your fruitful suggestion. We did accordingly and hope that the revised version is in better shape.

Comment 9.  Reference: Ref 50 do not have enough info such as volume, issue, page.  Furthermore, I unable to find the Ref 26, 27, 28 in the text. Please check.

Response to 9: Thanks for rising the point. We corrected Ref [50] and cited the references 26, 27, 28 in the text.

We are again very thankful to the reviewers and the editor for taking an interest in improving our manuscript and for suggestions and enhancing the manuscript. All the changes have been made in the article as directed. We hope that the article is now well in accordance.

Round 2

Reviewer 3 Report

The revise manuscript is acceptable